# Heat Transport Pathways into the Arctic and their Connections to Surface Air Temperatures

Daniel Mewes[1] and Christoph Jacobi[1]

[1]Leipzig Institute for Meteorology, University of Leipzig

**Correspondence:** Daniel Mewes (daniel.mewes@uni-leipzig.de)

**Abstract.** Arctic Amplification causes the meridional temperature gradient between middle and high latitudes to decrease. Through this decrease the large-scale circulation in the mid-latitudes may change and therefore the meridional transport of heat and moisture increases. This in turn may increase Arctic warming even further. To investigate patterns of Arctic temperature, horizontal fluxes and their changes in time, we analyzed ERA-Interim daily winter data of vertically integrated horizontal moist static energy transport using Self-Organizing Maps (SOM). Three general transport pathways have been identified: the North Atlantic Pathway with transport mainly over the northern Atlantic, the North Pacific Pathway with transport from the Pacific region, and the Siberian Pathway with transport towards the Arctic over the eastern Siberian region. Transports that originate from the North Pacific are connected with negative temperature anomalies over the central Arctic. These North Pacific Pathways are getting less frequent during the last decades. Patterns with origin of transport in Siberia are found to have no trend and show cold temperature anomalies north of Svalbard. It was found that transport patterns that favor transport through the North Atlantic into the central Arctic are connected with positive temperature anomalies over large regions of the Arctic. These temperature anomalies resemble the warm Arctic cold continents pattern. Further, it could be shown that transports through the North Atlantic are getting more frequent during the last decades.

## 1  Introduction

The Arctic regions play a significant and specific role in climate change. The temperature increases much faster compared to the rest of the globe (Stroeve et al., 2012; Wendisch et al., 2017), which is called Arctic Amplification. This stronger warming is mainly caused by loss of sea ice and the consequent increased exposure of the Arctic ocean to the atmosphere.

Following these changes in temperature and sea ice cover it was found that the sea level pressure (SLP) decreases over the Arctic in the winter season (Gillet et al., 2003; Screen et al., 2014). This itself might alter the circulation and thus the transport of air masses into and out of the Arctic. Analyses of the decadal variability in EC-Earth model (Hazeleger et al., 2012) runs showed that in a warmer climate the Aleutian Low intensifies during winter months, which changes the circulation patterns (Linden et al., 2017). The decrease of the temperature difference between the Arctic and mid latitudes due to Arctic Amplification is suggested to be followed by a change in the meridional transport of heat into the Arctic, which has been seen in reanalysis data (Graversen, 2006; Vinogradova, 2007). Analysis of regional climate model output has shown that at

the end of the 21st century the seasonal mean layer thickness between 1000 and 300 hPa over the Arctic will likely increase significantly, while the interannual variability increases (Rinke and Dethloff, 2008). To summarize, there is an indication that Arctic Amplification may alter the circulation and heat transport patterns in the Arctic.

To understand how circulation and transport are connected to other meteorological variables, the Self-Organizing Map
(SOM) method has been shown to be a viable cluster and pattern extraction tool (Liu et al., 2006; Liu and Weisberg, 2011). Cassano et al. (2006) evaluated model representations and projections of the SLP patterns over the Arctic. Corresponding temperature and precipitation anomalies have been attributed to the respective patterns that have emerged from SOM analyses. They found that SLP patterns that feature an extended North Atlantic storm track and a strong Aleuten low are connected with positive temperature anomalies. Negative precipitation anomalies over the North Atlantic were found for SLP patterns with
generally high SLP. Skific et al. (2009) used SOM analyses to validate performance of the Community Climate System Model. They showed that the model successfully captures major SLP patterns, which has been derived by the SOM from ERA-40 data. Additionally, they found through relating moisture transports to particular circulation regimes that by the late 21st century the transport is projected to be increased within the CCSM3. The SOM method was also used by Higgins and Cassano (2009) to determine the influence of reduced sea ice on the geopotential height of 1000 hPa over the Arctic using the CAM3 (Collins
et al., 2006). They found that with reduced sea ice the geopotential height of 1000 hPa increases over Siberia, the Greenland and Norwegian Seas.

Lynch et al. (2016) used the SOM method to evaluate the connection between SLP patterns and the connection with high and low sea ice cover in the Pacific sector. They showed that years with low ice fraction are connected with positive temperature anomalies and transport originating from the south, while years with high ice concentration are connected with transport of ice
from regions in the north even though the ice itself is melting.

Mattingly et al. (2016) have analyzed the tropospheric meridional moisture transport over Greenland using the SOM method and found that from 2000 to 2015 positive moisture transport anomaly patterns towards Greenland were more common compared to 1979 to 1994 and thus might have increased the melting of the Greenland ice sheet.

The question remains to which degree different heat transport pathways into the Arctic are responsible for the increased
Arctic warming. In this study we therefore focus on the moist static energy (MSE) transport pathways into the Arctic in the winter months and on the corresponding temperatures over the Arctic by using the SOM method. Winter was chosen as the Arctic temperature is most sensitive to influences through transports in this season (Yoshimori et al., 2017). We use ERA-Interim reanalyses (Dee et al., 2011; ECMWF, 2017) for clustering the vertically integrated MSE transport. Further, the temporal evolution of occurrence frequencies for the obtained patterns is analyzed, as well as corresponding temperature
anomalies for the Arctic region. In Sect. 2 the used data and SOM method are presented. The results are found in Sect. 3, and are followed by a discussion in Sect. 4. Section 5 concludes the paper.

## 2 Method and data

### 2.1 Data

In this study daily mean ERA-Interim (Dee et al., 2011) data were analyzed for the winter months (December to February) from
1979 to 2016. Data were provided at a horizontal grid resolution of $0.75° \times 0.75°$ on 60 vertical levels by ECMWF (2017). ERA Interim was chosen as it represents the temperature in the Arctic well (Chaudhuri et al., 2014; Simmons and Paul, 2015). Daily means were calculated from 6-hourly output. The vertically integrated daily horizontal MSE transport $\boldsymbol{MSET}$ is calculated at each grid point as follows:

$$\boldsymbol{MSET} = \frac{1}{g} \int_{\eta_{sfc}}^{\eta_{200\,hPa}} \boldsymbol{v}(\eta) \left(c_p\, T(\eta) + g\, z(\eta) + L\, q(\eta)\right) \frac{\partial p}{\partial \eta} d\eta. \tag{1}$$

Here, g is the gravitational acceleration (taken as 9.81 m/s$^2$), $\boldsymbol{v}$ is the horizontal wind vector, $c_p$ is the specific heat constant at constant pressure, $T$ is the temperature, $z$ is the geopontential height, L is the latent heat of vaporization of water, $q$ is the specific humidity, $\eta$ is the model level, and $p$ is the pressure. The integration limits from the surface to the $200\,hPa$ level were chosen to obtain the heat transport throughout the entire troposphere.

### 2.2 Self-Organizing Maps

The SOM is an artificial neural network developed by Kohonen (1998), and it is used to reduce the dimensionality of a data set by organizing it in a two-dimensional array, called map. SOMs were created by using the python package "somoclu" (Wittek et al., 2017). A general advantage is that the SOM is not limited by linear assumptions. Furthermore, the method shows advantages over PCA and (rotated) EOF analysis to find patterns in data (Reusch et al., 2005; Liu and Weisberg, 2011). SOMs are used as cluster analysis tool that broadly speaking is built to minimize the within cluster difference while maximizing the
between cluster difference. The SOM method is different in a way that neighbouring clusters within this two-dimensional map of clusters are more similar to each other than clusters that are farther apart from each other within the map. This is achieved by the characteristic of the SOM that the clusters also develop in dependence of the neighbouring clusters and thus retain the general topology of the multidimensional data it is used on. Eventually, each cluster represents a set of the given data. Thereby, the SOM as a whole is a representation of the data in a way that the emerging SOM shows more clusters that reproduce
topology and the distribution of the given data.

The SOM was used to analyze the tropospheric horizontal MSE transport calculated from ERA-Interim. The SOM clustering is used to find common transport features in the Arctic. Generally, the patterns emerges from the data due to the fact that the patterns try to minimize the within pattern variance. This is done by comparing each daily data field with each pattern and finding the pattern with the smallest Euclidean distance to the given daily data field.
Further, the two meter air temperature anomalies corresponding to the clustering of the tropospheric MSE transport are analyzed. This is done to obtain the respective transport effect on the temperature depending on the different transport features. As with a lot of other clustering algorithms, the choice of the right number of patterns to be extracted is partly subjective. A

SOM with the size of 4 columns × 3 rows was chosen for our analysis of MSE transport into the Arctic Ocean; it provided the best balance of generalization without loosing too many distinct states. In addition to the clustering of the SOM itself, we chose to group similar transport patterns that have emerged from the SOM manually.

Grouping the patterns is common in the literature (e.g. Mattingly et al., 2016; Higgins and Cassano, 2009). This serves to ease the discussion and provides the option to decide upon which patterns fit best not only based on the underlying Euclidean distances of the vectorized fields. The mathematical idea behind the manual grouping instead of using a smaller SOM size is that, due to the characteristics of Euclidean distance it is possible that with smaller SOM sizes distinct daily data fields might have been gathered in clusters that do not represent them well under meteorological point of view. The meteorological point of view in our case would be the general direction and rotations of the 2-dimensional vector field of the MSE transport. For other fields these might differ (e.g. for clustering temperature fields the meteorological point of view would be that slightly shifted maxima might cause big differences in the Euclidean distance) We decided to group them manually to make sure that patterns that fit under a meteorological point of view are gathered within a group, and thus share features that are more relatable to each other. SOM was chosen in favor of other techniques (e.g. k-means) because the patterns emerging from a SOM are easier to relate to each other by retaining the intrinsic topology of the data.

## 3 Results

### 3.1 Heat transport SOM

The SOM of the vertically integrated MSE transport is shown in Fig. 1. Each pattern features different transport strengths and directions as shown by the vectors. For a view on the general transport pathways we decided to further gather the patterns into three groups chosen according to the horizontal transport from middle latitudes into the Arctic Ocean. Thereby, we grouped patterns that show similar horizontal transport features to distinguish the respective composites for each of the found three major patterns. The manual grouping was based on the directions and rotations of the 2-dimensional vector fields. This manual grouping leads to a more transparent view on the actual clustering of the data.

The composite transports are shown in Fig. 2. They were derived by adding the distinct patterns of each group in Fig. 1 weighted by their relative frequency of occurrence. Subsequently, we will call them the North Atlantic Pathway, the Siberian Pathway, and the North Pacific Pathway.

The red framed patterns in Figs. 1 and 2 show the North Atlantic Pathway. Corresponding patterns for the North Atlantic Pathway are patterns 1.1, 1.2, 1.3, 2.2. These patterns share a heat transport that is going from North Atlantic either over Greenland or through the Fram Strait and over Svalbard into the central Arctic.

Patterns with a green frame correspond to transport that originates in central Siberia or northern Siberia and is directed into the central Arctic by a cyclone motion with its center over the Kara Sea, Laptev Sea, or the North Pole. These features are summarized as the Siberian Pathway. The Siberian Pathway consists of the patterns 2.1, 3.1, and 3.2. The transport structure of Pattern 2.1 is mainly zonal within the central Arctic, and no strong meridional transport is present. However, its general structure with a centralized center of cyclonic motion fits best into the Siberian Pathway group.

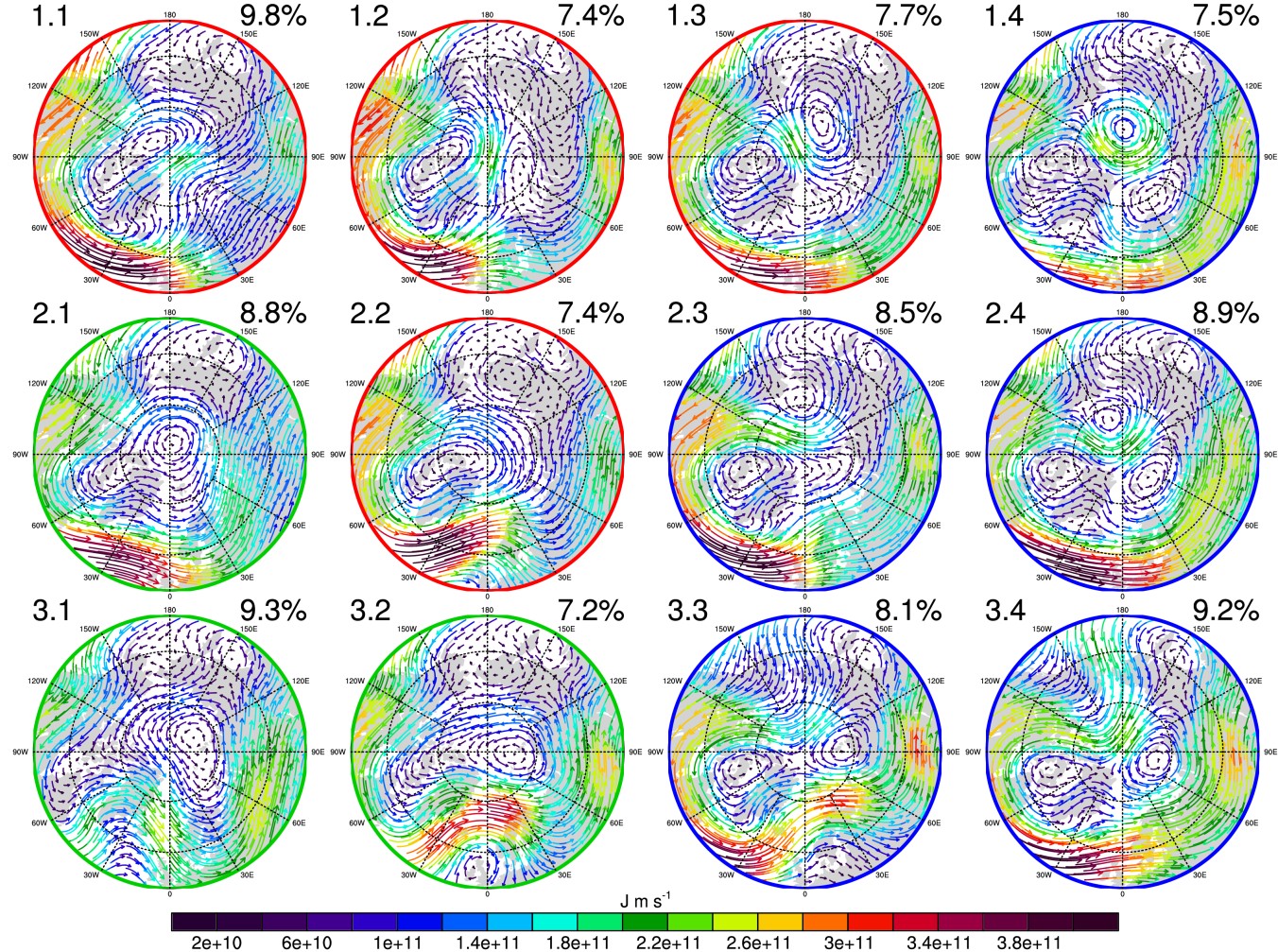

**Figure 1.** 4×3 SOM of vertically integrated (200 hPa -1000 hPa) horizontal MSE transport from winter ERA-INTERIM data (1979-2016). Numbers on the top left are used to name different patterns, percentages in the top right of each pattern correspond to the relative frequency of occurrence during the analyzed time period. The maps are centered at 0°E. Red vectors correspond to stronger transports, while blue vectors correspond to weaker transports. Differently colored frames indicate patterns that were grouped together.

The North Pacific Pathway (blue frames) arises from the patterns 1.4, 2.3, 2.4, 3.3,and 3.4. The main transport occurs from the North Pacific through east Siberia into the central Arctic. This occurs mostly with one center of counter-clockwise transports at the Barents Sea or Laptev Sea and the other center over the Northwest Passage. In some cases (see patterns 1.4, 2.3, and 2.4) clockwise transports with the center north of the Bering Strait or within the central Arctic are present.

Due to the grouping the mean within cluster variance has changed from $2.2 10^{22}$ J m s$^{-1}$ for the twelve SOM clusters to $2.7 10^{22}$ J m s$^{-1}$ for the three pathways.

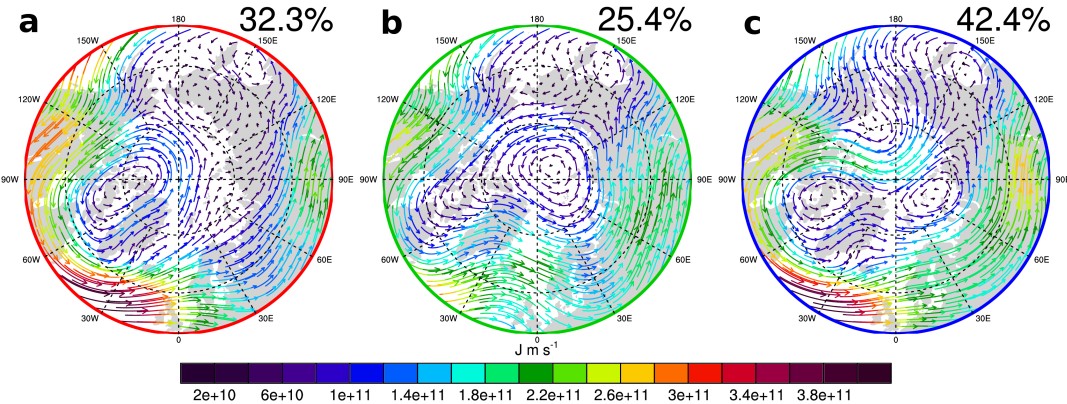

**Figure 2.** The three different transport pathways: the North Atlantic Pathway (a; red colored frame), the Siberian Pathway (b; green colored frame), and the North Pacific Pathway (c; blue colored frame), derived by the composites of the selected patterns of Fig. 1 weighted by their relative frequency of occurrence within the group.

## 3.2 Temperature anomaly composites according to transport pathways

Figure 3 shows the composites of the temperature anomalies corresponding to the respective pathways. Temperature anomalies were calculated as deviations from the winter mean period from 1979 to 2016. The red framed plot (Fig. 3a) shows the anomalies related to the North Atlantic Pathway, the green one (Fig. 3b) those related to the Siberian Pathway, and the blue one (Fig. 3c) those related to the North Pacific Pathway. The North Atlantic Pathway related temperature anomalies (Fig. 3a) show

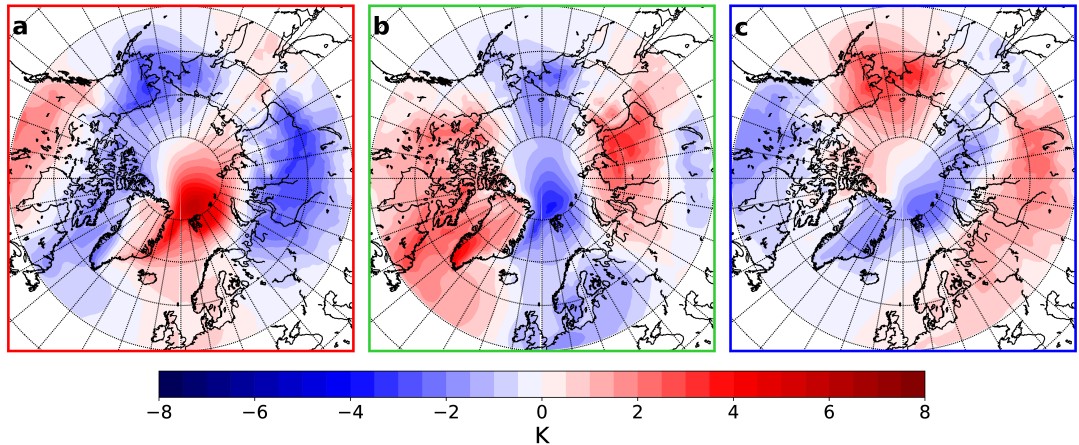

**Figure 3.** Composite of 2 meter temperature anomalies for each of three the pathways: the North Atlantic Pathway (a; red colored frame), the Siberian Pathway (b; green colored frame), and the Northern Pacific Pathway (c; blue colored frame). Contour spacings show temperature anomalies in 0.5 K. Blue colors indicate a cold anomaly and red colors indicate a warm anomaly compared to the mean of the analyzed time frame.

increased temperature from the North Atlantic into the central Arctic with a maximum greater than 6 K north of Svalbard. For northern Canada, the Bering Strait and central Siberia a cold anomaly is observed with a minimum of $-3.5$ K at the Bering Strait and north of Lake Baikal. The negative anomaly in Siberia results from the increased transport over the North Atlantic, which results in a decrease of zonal transport of MSE to Siberia, and in an increase of transport of cold air from the north.

The Siberian Pathway (Fig. 3b) is connected with higher temperatures over Siberia, as well as with warm anomalies over Northern America and Greenland with temperature anomalies greater than 5 K at the southern tip of Greenland. Negative temperature anomalies occur over Northern Europe, through the Fram Strait and Svalbard into the central Arctic, the Chukchi Sea, and the Bering Strait, with anomalies as low as $-4$ K north of Svalbard. This temperature pattern occurs because of the limited MSE transport through the North Atlantic and the more zonally favored transport over Europe and Siberia.

The North Pacific Pathway (Fig. 3c) composite shows increased temperature over large parts of Eurasia connected with zonal transport over the continent. Positive temperature anomalies are also seen over the Bering Strait, and the Chukchi Sea (up to 4 K), together with northward transport there (Fig. 2c). From North America over Greenland and Svalbard to the Laptev Sea a cooling effect is observed, with the maximum of $-3$ K west of Svalbard.

Figure 4 shows the composites of the vertically averaged (surface to 200 hPa) potential temperature anomalies corresponding to the respective pathways. As vertically integrated transports are analyzed, the vertical averaged potential temperature shall provide a more relatable quantity compared to the 2 meter temperature. Generally, the vertically averaged potential temper-

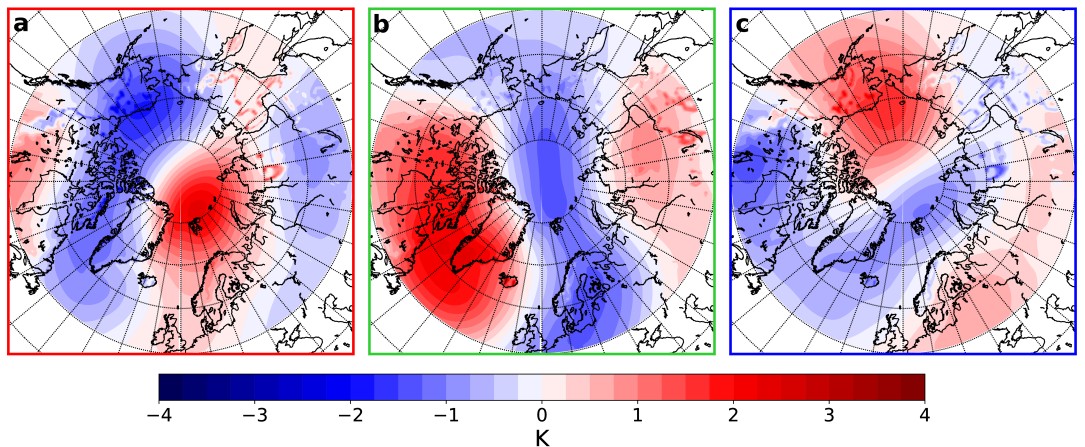

**Figure 4.** Composite of vertically averaged (surface to 200 hPA) potential temperature minus mean for the winter mean period from 1979 to 2016

ature anomalies show very similar structures compared to those of the 2 meter temperatures for all pathways. The general maximum amplitude of the vertically averaged potential temperature anomalies is smaller by a factor of two. This shows that the corresponding 2 meter temperature anomaly fields for each pathway are a good indicator of the vertically averaged potential temperature within the troposphere. However, the specific locations of maxima and minima is slightly shifted for the

Siberian Pathway, which does not show a clear negative anomaly north of Svalbard but a elongated region of negative anomaly of potential temperature.

## 3.3 Meridional heat transport

In order to more clearly show the transport of MSE into the Arctic through the three identified patterns, we analyzed the longitudinal distribution of the meridional component of equation 1.

The mean of the meridional transport at $75°$ N is shown in Fig. 5.

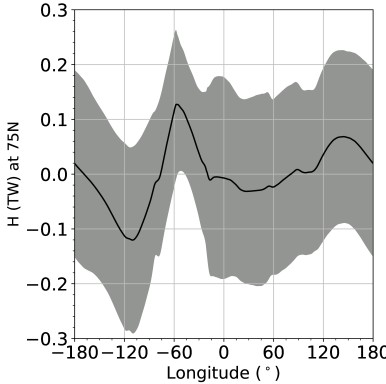

**Figure 5.** Mean vertically integrated (200 hPa-1000 hPa) meridional MSE transport at $75°$ N given in Terawatts. Grey shaded areas show the standard deviation based on the daily data.

Figure 6 shows the meridional MSE transport anomalies at $75°$ N at each longitude grouped into the three transport pathways. The amplitude of the respective meridional transports are a measure of the general energy content (compare equation 1); No
advection (transport across the gradient) is shown. Note that the standard deviation is in the order of 0.2 TW due to the used daily data of the corresponding group that is highly variable. For the composite of the North Atlantic Pathway (red, Fig. 6a) we have maximum positive anomalies of the meridional MSE transport from $50°$ W to $50°$ E, and negative anomalies from $60°$ E to $140°$ W. The North Pacific Pathway (blue, Fig. 6c) is connected with positive transport anomalies from $80°$ E to $170°$ W. This corresponds to the described pathway: originating from the North Pacific and going over Eastern Russia to the central
Arctic. The Siberian Pathway (green, Fig. 6b) shows positive anomalies from $180°$ W to $60°$ W and from $30°$ E to $100°$ E.

## 3.4 Trend of transport pathways

Overall, the North Atlantic pathway occurs during about 32 %, the North Pacific pathway during about 42 %, and the Siberian Pathway during about 25 % of the analyzed time period. For each of the three groups the relative frequency of occurrence was calculated for each winter and the respective time series are shown in Fig. 7. A positive trend has been found for the North
Atlantic Pathway (Fig. 7a), a negative trend for the North Pacific Pathway (Fig. 7c), and no trend for the Siberian Pathway (Fig. 7b). Note that the trends for each of the three patterns are not independent from each other. We also note that the positive

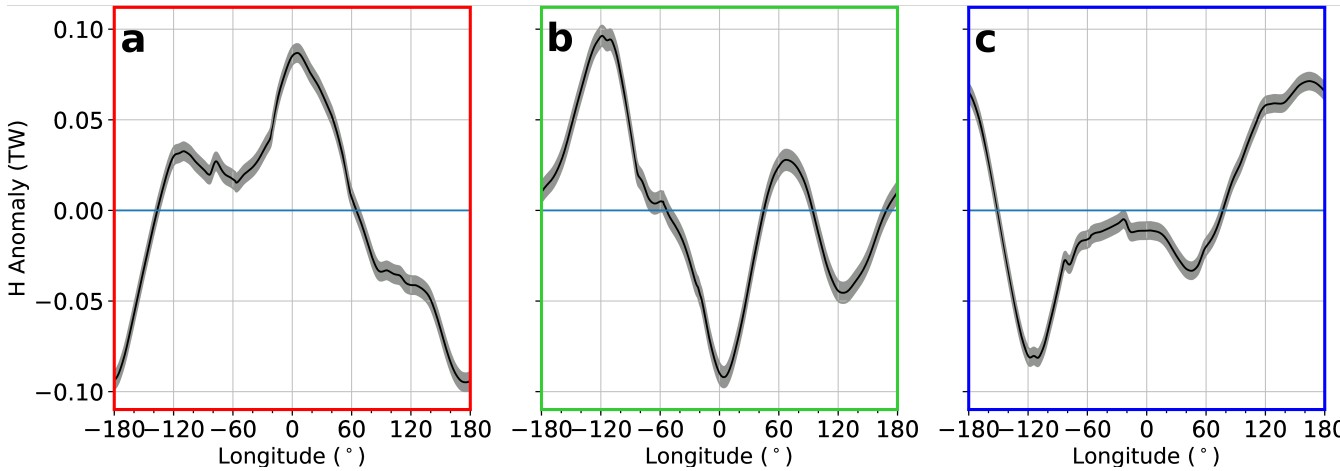

**Figure 6.** Composite of meridional MSE transport anomalies at 75° N given in TW for each of the three pathways: (a) North Atlantic Pathway, (b) Siberian Pathway, (c) North Pacific Pathway. The mean values have been shown in Fig. 5. Positive sign denotes a transport anomaly to the north and negative sign denotes a transport anomaly to the south.

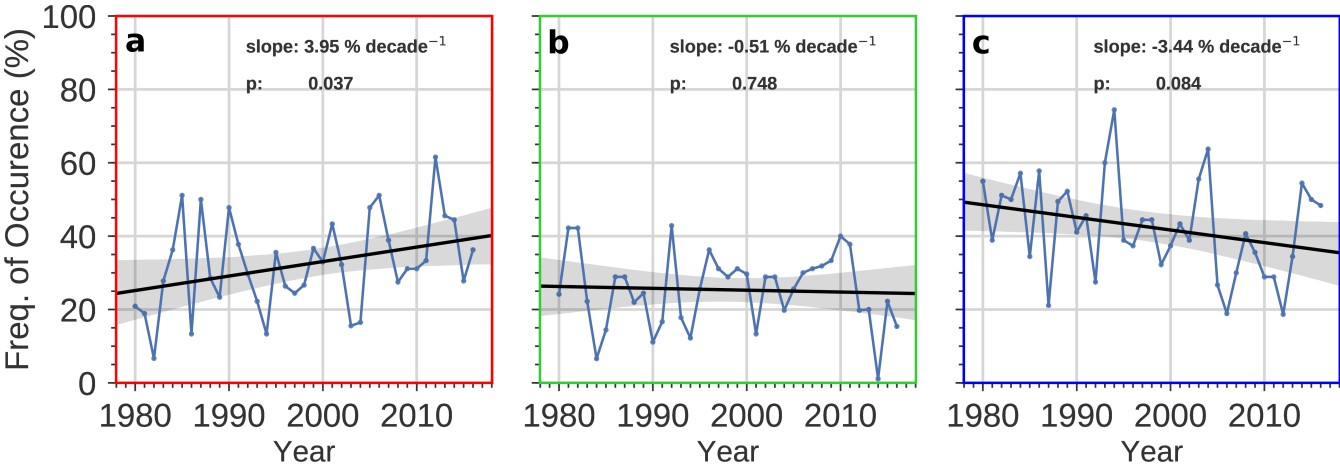

**Figure 7.** Frequency of occurrence for each transport pathway group according to the coloring (Fig. 1). The blue line shows the frequency of occurrence for each years winter from 1979 to 2015. The black line shows the linear trend line. Greyshading shows the 95 % confidence intervals for the trends derived via bootstrap re-sampling. p values according to a 2 sided t-test are shown in the respective panels.

and negative trends shown in Fig. 7a and Fig. 7c are not robust, and there is a small probability that they might indeed be different than derived from the linear fit.

## 3.5 Temperature trends

Comparing the general temperature trend with the resulting temperature anomalies due to different transport pathways indicates to which degree heat transports might play a role for the warming of the Arctic. The general temperature trends for the winter season during the analyzed time period is shown in Fig. 8a. The trends were derived through a Theil-Sen regression, which is robust against outliers. Generally it can be seen that the trend exceeds $3.5\,\mathrm{K\,decade^{-1}}$ for the region east of Svalbard. Largest

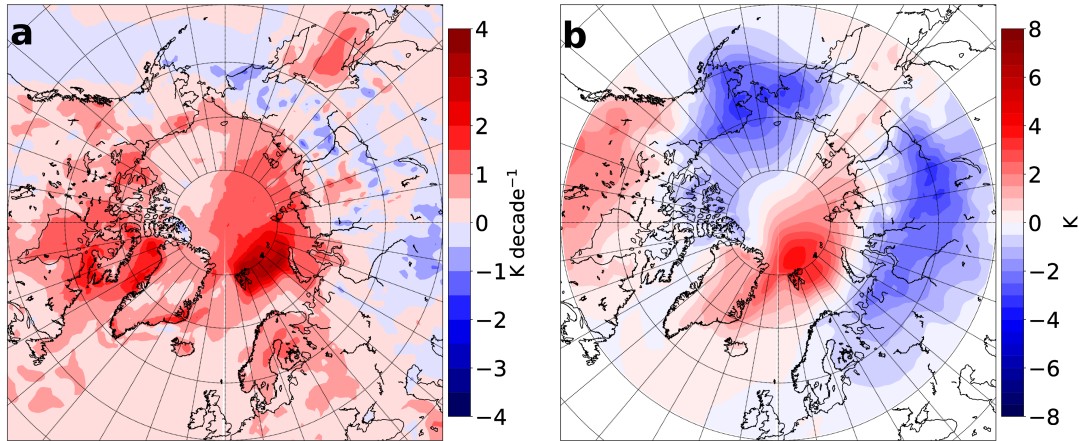

**Figure 8.** (a) temperature trends for the winter mean, calculated for the winters from 1979/80 to 2015/16. Trends are given in Kelvin per decade. (b) composite of temperature anomalies from the North Atlantic Pathway and the negative of the North Pacific Pathway temperature anomalies as provided in Fig. 3, weighted by their relative frequency of occurrence (Fig. 2 top right numbers).

positive trends are found for regions of the Barents Sea, the Kara Sea, the Laptev Sea, and the Baffin Bay. Negative temperature trends occur over Siberia, but only for very small regions.

To calculate the influence of changes in transport pathways we calculated the weighted average of the temperature anomaly of the North Atlantic Pathway and the negative of the temperature anomaly of the North Pacific Pathway (Fig. 8b). This was done to take into account the influence of an increased occurrence frequency of the North Pacific Pathway and a decrease of occurrence frequency for the North Pacific Pathway, and thus to analyze the possible change in temperature according to a trend in the transport pathways. Each of the temperature anomaly fields were weighted by the relative frequency of occurrence shown in Fig. 2. The Siberian Pathway was not included as it does not show a trend in the occurrence frequency. This new composite shows similar features compared to the temperature anomaly of the North Atlantic Pathway (Fig. 3a), which is owing to the fact that the temperature anomalies connected to the North Atlantic and North Pacific Pathways are broadly inverse to each other.

The regions of large temperature anomalies are more confined and weaker then the ones considering single pathways alone. The largest positive temperature anomaly occurs north of Svalbard with up to 3.5 K. Negative anomalies occur over the Bering Strait (-3.0 K) and north of Lake Baikal (-2.5 K).

The winter temperature trend shows a strong positive signal east of Svalbard. This signal can partly be seen in the temperature anomaly which also shows a positive signal in this region. For the regions that correspond to lower temperatures with an increased occurrence of the North Atlantic Pathway and a decreased occurrence of the North Pacific Pathway no uniform temperature trend can be found on Fig. 8a. This suggests that the temperature anomalies due to the transport changes are counteracted by other processes. It has to be noted, that the MSE transport cannot be accounted for changes in the temperature anomalies alone. Other transports and processes affect the temperature as well, even to a higher degree.

## 4 Discussion

The change of influence and connection of atmospheric circulation with surface temperature is a highly discussed topic, especially in terms of the increased temperature rise in the Arctic. Here we grouped data according to distinct pathways based on SOM analysis and looked at related temperatures and the respective trend of the pathways.

It has to be noted that the linkage between the analyzed vertically integrated MSE and 2 meter temperature anomalies are not as straight-forward. The general connection between the shown MSE fluxes and the 2 meter temperature is due to the convergence and divergence of the flux which did not show good agreement between the two quantities. The strongest influence of the energy transports on the surface air temperature is delayed by about 5 days (Graversen, 2006), but in this study we did not consider the persistence of each pattern. However, within this study the focus was to connect the distinct pathways to the concurrent 2 meter temperatures and to get an idea of the conditions during distinct pathways.

The increase in frequency of transports through the Northern Atlantic, as shown for the North Atlantic Pathway, has also been found by Dahlke and Maturilli (2017). They analyzed the transports of air masses to the region of Ny-Alesund using backward trajectories. They were able to find a more frequent source region of air masses in the North Atlantic, while we could show that and the transport through the North Atlantic is getting more frequent. Dahlke and Maturilli (2017) identified a positive temperature anomaly over the Svalbard region that is connected with changes in advection of air masses. We find that the increased frequency of the North Atlantic Pathway is connected with temperature anomalies that favor a strongly positive anomaly in the central Arctic and strongly negative anomalies over Siberia and the Bering Strait. This is following from the transport of heat to the northern regions instead of transport to Siberia.

The vertically averaged potential temperature composites are connected to the 2 meter temperature anomalies. However, the potential temperature anomalies show a more general state of the troposphere, but generally show nearly identical features as the 2 meter temperature anomaly composite.

The 2 meter temperature composite from the North Atlantic Pathway has also similar features compared to the cold continents and warm Arctic proposed by Overland et al. (2011). In our analysis negative temperatures anomalies over Canada are not seen. But the cold anomalies over central Siberia, as well as the warm anomaly sector over the central Arctic are quite well reproduced for transports through the North Atlantic.

Adams et al. (2000) found transport of heat from the North Atlantic and North Pacific to the Arctic for transient and stationary eddies. Also Messori et al. (2018) found a systematic transport of moisture through the Atlantic sector into the Arctic for warm

spells. These warm spells are accompanied by advection of cold air across Siberia, which can be partly seen in the temperature composite of our the North Atlantic Pathway. The transports into the Arctic discussed by Messori et al. (2018) are comparable with the transports shown in our results. The general trend of increased northward transport of air can also be seen in regional analysis by Mattingly et al. (2016). They focused on the moisture transport over Greenland. Their analysis shows an increase of moist states over Greenland, which are partly connected with more northward transports. Rinke et al. (2017) analyzed extreme cyclone events in the Arctic wintertime from measurements at Ny-Alesund and from ERA-Interim reanalysis. They found that the number of extreme cyclone events increases. For days with extreme cyclone events at Ny-Alesund their temperature anomaly pattern looks similar as the temperature pattern shown in the North Atlantic sector for the North Atlantic Pathway. This suggests that the origin of the extreme cyclones analyzed in Rinke et al. (2017) might be connected with increased transport through the North Atlantic sector to Svalbard.

Woods et al. (2013) analyzed poleward moisture intrusions across 70°N for winter months using ERA-Interim reanalyses. The concentration of these intrusions were found to be at latitudinal regions of the Labrador Sea, the North Atlantic, and the Barents/Kara Sea and the Pacific. These regions are partly presented by the pathways shown in this work: the general intrusions through the Atlantic and Pacific are captured by the North Atlantic and North Pacific Pathway. Intrusions through the Barents/Kara Sea seem to be captured also by the North Pacific Pathway, while the intrusions through the Labrador Sea cannot be distinguished easily within the pathways. However, we considered MSE transport instead of the latent heat transport only. Specifically for December and January Woods and Caballero (2016) could show a positive trend of the total number of intrusions and their connection to surface air temperature and sea ice cover. Largest influences in temperature were observed over the Barents Sea. They showed that the intrusions show typical directions from the North Atlantic into the Barents Sea. The Barents Sea region shows positive temperature anomalies for the North Atlantic Pathway, which features a positive trend. These connections are in agreement with the discussed literature.

For the winter months, Cassano et al. (2006) were able to connect SLP patterns with lower pressure over the Bering Strait and the North Atlantic and higher pressure over Siberia with a temperature anomaly that shares very similar features to those of the North Atlantic Pathway found in the work presented here.

The frequency of occurrence of the North Pacific Pathway decreases during the last decades. This is connected with less frequent negative temperature anomalies over the central Arctic. For the winter Matthes et al. (2015) show that the number of cold spell events is decreasing over Scandinavia and Northern Canada, while for Siberia also regions with an increase of cold spells could been found. Warm spells showed strong significant increase over Scandinavia. Matthes et al. (2015) analyzed the trends for warm and cold spells over the land masses and islands in the Arctic using daily station data and ERA-Interim reanalysis. Looking at the trend of regional temperature extremes at Ny-Alesund, Wei et al. (2015) could show that cold extremes have a negative trend and warm extremes have a positive trend. These results agree with the connection of the North Pacific Pathway (North Atlantic Pathway) to cold (warm) temperature anomalies and a decrease (increase) in frequency of occurrence.

We compared the resulting mean temperature anomalies for the general change in transport – decrease of occurrence frequency of North Pacific Pathway and increase of occurrence frequency of North Atlantic Pathway – with the general temper-

ature trend for the winter season from 1979/80 to 2015/16. We found trends over $3.5\,\text{K}\,\text{decade}^{-1}$ for the general temperature trend in winter west of Svalbard. Graversen (2006) analyzed the influence of the atmospheric northward energy transport on the surface air temperature for ERA-40 reanalysis for the years 1958 to 2001. He found that the atmospheric northward energy transport addresses about $0.15\,\text{K}\,\text{decade}^{-1}$ over Svalbard. Compared to our analyzed time frame this would add up to about $0.6\,\text{K}$ anomaly over Svalbard. We identified a positive temperature anomaly of about $3.0\,\text{K}$ over Svalbard, which is about $2.4\,\text{K}$ more than explained by the total atmospheric northward energy transport. Due to finding connected temperature fields for distinct transport pathways, we are able to see all influences of the atmosphere under these specific pathways and not only the specific influence of the northward energy transport, which was analyzed by Graversen (2006).

It was found that in regions where the change in transport will favor negative temperature anomalies (Siberia and Bering Strait) the temperature trend is not as uniform. For regions north and east of Svalbard the change in transport is connected with positive temperature anomalies that also coincide with regions of positive trends in temperature. Comparing the combined composite of temperature anomalies connected to the changes in the major transport pathways (Fig. 8b) to the temperature anomalies of the Siberian Pathway (Fig. 3b) shows that in general the central Arctic tends to become warmer while the Bering Strait tends to become cooler in relation to the change in transport. So in general, the change of transports would lead to more frequent negative temperature anomalies over the Bering Strait and Siberia. These cannot be seen in the trends shown on Fig. 8a. Our results show the expected geographic distribution of surface temperature anomalies that coincides with theses changes in the transport. These results are also a good example that the surface trend is influenced by a lot of processes and cannot be discussed solely by heat transport alone.

## 5 Summary and conclusion

With the SOM method we were able to find intrinsic MSE transport patterns within the MSE transport fields and used them as a guide for our analysis. Three distinct transport pathways were extracted from the SOM analysis: the North Atlantic Pathway, the Siberian Pathway, and the North Pacific Pathway. The North Atlantic Pathway is connected with with transports through the North Atlantic into the Arctic, the North Pacific Pathway is connected with transports that originate from the North Pacific and enter the Arctic through east Siberia, and the Siberia Pathway is features by transports through the Arctic from central Siberia. We analyzed the temperature anomalies that are related to the different transport pathways. This type of analysis helps to get a more complete view of the atmosphere during these different transport pathways.

We conclude that during the last decades the transport through the North Atlantic into the Arctic has increased. These North Atlantic Pathways are connected with positive temperature anomalies over the Arctic, and negative temperature anomalies over the Bering Strait and central Siberia. This shows that relating temperature anomalies based on the transport alone is favouring an increased pattern of warm Arctic and cold continents. Thus it can be stated that the warm Arctic and cold continents pattern is partly controlled by the increased northward MSE transport through the North Atlantic.

A question that still remains open is the question of causality. To which degree the change in MSE transports and circulation is changing the temperatures in a warming Arctic and to which degree is the temperature change influencing the heat transports and circulation themselves cannot be decided based on SOM analysis alone.

5   *Author contributions.* Daniel Mewes performed the data analysis and wrote the first draft of the manuscript. Christoph Jacobi initiated the project and made contributions to the results interpretation and manuscript writing.

*Competing interests.* The authors declare that they have no conflict of interest.

*Acknowledgements.* ERA-Interim reanalyses data have been provided by ECMWF apps.ecmwf.int/datasets/data/. We gratefully acknowledge the funding by the Deutsche Forschungsgemeinschaft (DFG, German Research Foundation) – Projektnummer 268020496 – TRR 172,
10  within the Transregional Collaborative Research Center"ArctiC Amplification: Climate Relevant Atmospheric and SurfaCe Processes, and Feedback Mechanisms (AC)[3]" in the sub-project D01. We thank Annette Rinke, AWI Potsdam, for helpful discussions and comments. We want to thank the anonymous reviewer and Rodrigo Caballero for the helpful comments that very much helped us to improve the paper.

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
