# Peer review of "Heat Transport Pathways into the Arctic and their Connections to Surface Air Temperatures"

_Atmospheric Chemistry and Physics, 2018_

## Referee Comment (RC1) · Anonymous Referee #1 · 7 Sep 2018

Review of ACPD-2018-636

**"Heat Transport Pathways into the Arctic and their Connections to Surface Air Temperatures"**

by D. Mewes and C. Jacobi

This paper addresses the wintertime transport of heat into the Arctic and aims at identifying the major pathways and trends in those. For that purpose the authors apply the method of Self-Organizing-Maps to vertically integrated horizontal heat transport derived from the ERA-Interim reanalysis. They identify three preferential large-scale patterns of heat transport over the Northern Hemisphere mid- and high latitudes. Trends in the frequency of these transport patterns are then related to warming trends in the Arctic. Specifically, their analysis suggests that the transport through the North Atlantic sector has become stronger throughout the analysis period.

Given the amplified warming of the Arctic, especially in the European high Arctic, the topic of this paper is highly relevant. Furthermore, the approach to use Self-Organizing-Maps for identifying the dominant atmospheric patterns of heat transport – though not novel to the stratification of atmospheric data – is certainly innovative. The study suffers, however, from several methodological issues as outlined below. These issues need to be resolved before this paper can be considered publishable. Furthermore, the overview of the existing literature is somewhat on the short side and should be extended. Overall, my recommendation is major revisions.

**Specific comments:**

1.  I am puzzled that there is such a strong heat transport over Greenland in many of the clusters (most pronounced in the cluster 2.2, 3.1 and 3.2). The transport clearly seems to increase there. I would expect the opposite since the vertical integral of the heat transport is proportional to the thickness of the layer from the surface to the 200 hPa level such that over high topography it should decrease. The same concerns intense transport over the Tibetean plateau and across the Rockies in some clusters.

    I strongly suspect that the authors used data on the default pressure levels for computing the vertically integrated heat transport and they did not take into account that some pressure levels intersect the surface. Over topography, where surface pressure can be below the pressure of, for example the 1000 hPa level, the ECMWF provides extrapolated fields. The heat flux computed from such extrapolated fields is unphysical. This suspicion is further fuelled by the fact that there are no substantial T2m anomalies over Greenland in clusters with a strong heat transport there.

    The method section does not provide evidence of how the computation is done and what type of levels are used. It only states that the data is available on 37 levels. The default set of pressure levels comprises 37 levels, whereof only 23 are located between 1000 hPa and 200 hPa. Anyway, the above should be clarified and if my suspicion is true,

the computation of the vertically integrated heat flux needs to be redone properly. Otherwise the reason for the high heat transports above high topography must be explained.

2. The authors decide to perform a SOM analysis using 4x3 patterns. The number of patterns needs to be prescribed and is a subjective choice required by this method, which as such is fine. What I am worried about, however, is that they subsequently lump several clusters together according to the heat transport at 80°N, resulting in three major patterns, which are then considered for the rest of the paper. The only argument provided to justify this approach is that "this is common practice in SOM analysis". In my view, the SOM analysis on 2d fields is an overkill to obtain just these three patterns that are based on 1d fluxes at 80°N. If mainly the transport at 80°N is of interest for stratifying the patterns, why not perform an EOF or clustering analysis on the transport at 80°N and start from there?

To make the effort put into the SOM analysis more valuable, I'd suggest to present some additional analyses that make use of the nuances displayed in the 12 SOM clusters. For example, the authors could display other composited fields for these clusters. These fields could, for example, be T2m anomalies, and frequencies of atmospheric weather systems (cyclones, blockings). The latter would allow to relate the heat transports patterns to the dynamics, which I think would be a valuable addition and strengthen the paper.

3. There is a misleading use of statistical significance testing at several occasions in the paper. Statistical significance testing provides information about the likelihood of e.g. observing a certain trend under the assumption of the null hypothesis (the trend is not real and just the result of the sampling). It does not provide any evidence about whether the trend is real. A large p-value simply implies that the data is consistent with the null hypothesis and there is no evidence for a trend. It does also not rule out that there is a trend, but given the data we cannot tell. A low p-value in turn does not indicate a high likelihood for the hypothesis to be true, it just tells us that the data is unlikely to be observed under the assumption that the null hypothesis is true (for a thorough discussion see Ambaum 2010: Significance test in climate science, J. Climate, 23, 5927 – 5932).

Specifically in this paper the following misleading use of significance testing occurs:

○ P8 L6: "...the group of the Siberian pathway does not provide a significant trend..." This is a misleading statement, as it suggests that we should trust the weak trend in the frequency of the Siberian pathway less than the other trends, which we should trust because they are significant. This is of course absurd because the sum of the trends needs to balance. Hence, if we trust the trends of the other two patterns, we have to trust the weak trend of the third by as much. The significance test is therefore not helpful.

It would be more insightful to provide confidence intervals that illustrate the robustness of the trends to rule out that the trend is strongly influenced by a few data points.

○ P9 L11: "For the regions that correspond to lower temperatures with an increased occurrence of the North Atlantic Pathway and a decreased occurrence of the North Pacific Pathway no significant temperature trend can be found on the left panel of Fig. 7. This suggests that the temperature anomalies due to the transport changes are counteracted by other processes."

The reasoning is wrong here. It could well be that there is a cooling trend at these locations, but the trend is hidden because of one or two much warmer winters, which may be outliers. The statistical significance test does not provide us any information either way. A Monte Carlo resampling assessing the robustness of the local trends would give more insight.

○ In Fig. 7 (left panel): The significance test here does again not provide any information about whether the trends in some regions are robust. Again confidence intervals would be more insightful. Furthermore, in multiple testing scenarios, if any significance test is done at all, a field significance test should be done to take spatial correlations and erronous rejections of the null hypothesis into account (cf. Ventura et al. 2004: Controlling the Proportion of Falsely Rejected Hypotheses when Conducting Multiple Tests with Climatological Data, J. Climate, 17, 4343 - 4356)

4. What is the reason for not considering moisture fluxes as well? These are arguably highly important for Arctic heat anomalies because of their impact on the radiation balance. See for example Woods et al. (2013) and Messori et al. (2018; cited in the paper).

Woods, C., R. Caballero, and G. Svensson (2013), Large-scale circulation associated with moisture intrusions into the Arctic during winter, Geophys. Res. Lett.,40, 4717–4721, doi:10.1002/grl.50912.

5. For displaying the differences between individual clusters more clearly, it could help to show heat transport anomalies instead of the full transports. In many patterns the differences in the heat transport are rather nuanced and hard to see.

6. How large is the within cluster variance for the SOM clusters and the three main clusters?

7. P5 L10ff: T2m anomalies: Why don't you consider vertically averaged / integrated (potential) temperature anomalies? These would be more clearly related to the heat flux divergence than T2m anomalies, which are strongly influenced by surface heat fluxes. This is especially true in regions with a rapidly declining wintertime sea ice cover

(Barents and Kara Seas), where the temperature trends are to a large extent due to surface heat fluxes.

8. Related to the above I would be interested in seeing the divergence of the heat flux. Warming at a certain location will be more related to the heat flux divergence than the flux itself.

9. P5 L10ff: How are anomalies computed? Are they taken from the period (DJF 1979 – 2016) mean or is a running mean used to account for intra-seasonal variations?

10. Fig. 4: There seems to be a large compensation between the poleward and the equatorward transports, which I find surprising, especially concerning the strong southward heat transport at -120°E, which must be associated with very cold air (with low heat content).

    And does the standard deviation depict the inter-annual variability? That is, is it computed from the means of each winter? Or is it the standard deviation computed from daily data?

11. Fig. 7 (right panel): The caption should state that for the North Pacific pathway the inverse of the temperature anomaly was taken (as described in the text).

12. P1 L16: To first order the much stronger warming in the Arctic compared to lower latitudes is caused by the loss of sea ice, exposing major areas of the Arctic ocean to the atmosphere, leading to subsequent warming of the lower troposphere, and not the other way round. Additional melting of sea ice because of Arctic amplification would require additional transport of heat into the Arctic.

13. P2 L1: "To summarize, there is a clear indication that Arctic Amplification alters the circulation and heat transport patterns in the Arctic." I'd suggest to tone this statement down a bit. The causality is not fully clear in my view. See also Screen et al. 2018: Consistency and discrepancy in the atmospheric response to Arctic sea-ice loss across climate models, Nat. Geosci., 11, 155 - 164

14. P2 L6: "… that have been emerged …" → "… that have emerged …"
15. P2 L9: pattern should be patterns
16. P2 L9: either high or strong, not both
17. P2 L26: reanalyses → reanalysis
18. P2 L28: Please rephrase "This is used to obtain informations from the whole tropospheric column."
19. P3 L11: "… an average picture …" → the heat transport throughout the entire troposphere (?)
20. P4 L4: Fig. → Figs.
21. P4L6: This is likely an artefact from the vertical averaging

22. P5 L1: … are directed → … is directed …
23. P5 L4: zonally → zonal
24. P5 L7: Awkward formulation " … with two cyclone motions …", please rephrase
25. P5 L9: "… an anti-cyclone motion…" dito
26. P6 L12: Why focus on 75°N when SOM clusters are grouped together according to the heat flux at 80°N? Generally, I think 75°N is better suited because 80°N lies largely in the interior Arctic (except for the European sector).
27. P6 L33: Since you integrate H vertically, you could simply state that you consider the meridional component of the heat flux Eq. (1).
28. P7 L4: Remove "Generally, the meridional transports of the three groups fit well to the described pathways." - of course they have to be consistent as you look at the same quantity (the vertically integrated heat transport).
29. P8 L2: occur → occurs
30. P9 L14: remove "can not"
31. P10 L6: favors → favor
32. P10 L20: measurement → measurements
33. P10 L25: Awkward phrasing, please rephrase.
34. P10 L28: the presented work here → the work presented here
35. P10 L31: a increase → an increase
36. P11 L3: generally → general
37. P11 L13: that at region → that in regions
38. P11 L16: changing of → changes in
39. P11 L20ff: awkward phrasing until and including "… whole picture."
40. P11L27: an guide → a guide

---

## Referee Comment (RC2) · R. Caballero (Referee) · 14 Sep 2018

Review of "Heat Transport Pathways into the Arctic and their Connections to Surface Air Temperatures" by D. Mewes, C. Jacobi.

Overview: This paper applies the self-organized maps algorithm to cluster maps of sensible heat transport into preferred types and relate them to surface temperature anomalies and trends in the Arctic. This could in principle be of some interest. However, I think that the manuscript as it stands gives too little insight into the underlying physical mechanisms and it is difficult to see how it contributes to the current debate regarding Arctic warming and polar amplification.

[Figure]

Main comments:

1. The study claims to study "heat transport", but actually only studies one component of the heat transport. The relevant quantity for atmospheric energy transport is the moist static energy, $h = c_p T + g z + L_v q$ (where $g$ is gravitational acceleration, $z$ geopotential height, $L_v$ latent heat of vaporization and $q$ specific humidity). The authors only consider the first term, and neglect the others for no clear reason. In fact, recent work (see references below) shows that the latent heat component (i.e. the moisture transport) is the most important for warming the surface in the Arctic. The authors should cite these papers. Even the Yoshimori etal paper, which is cited by the authors, makes this point very clearly. The fact that moisture transport is not considered makes physical interpretation of the authors' results difficult – it's not clear if there is any direct causality implied by the relation between sensible heat transport and surface temperature anomalies shown here. It is thus not clear to me how this paper contributes to the current debate about Arctic warming. To make a clear and useful contribution, the authors really would need to apply their SOM classification to moisture transport and assess the pathways they obtain. It would also be useful to do a classification for dry static energy ($c_p + g z$) transport.

[Lee et al., 2017] Lee, S., Gong, T., Feldstein, S. B., Screen, J. A., and Simmonds, I. (2017). Revisiting the cause of the 1989–2009 Arctic surface warming using the surface energy budget: Downward infrared radiation dominates the surface fluxes. Geophys. Res. Lett., 44:10,654–10,661.

[Park et al., 2015a] Park, D.-S. R., Lee, S., and Feldstein, S. B. (2015a). Attribution of the recent winter sea ice decline over the Atlantic sector of the Arctic Ocean. J. Climate, 28:4027–4033.

[Park et al., 2015b] Park, H.-S., Lee, S., Kosaka, Y., Son, S.-W., and Kim, S.-W. (2015b). The impact of Arctic winter infrared radiation on early summer sea ice. J. Climate, 28:6281–6296.

[Park et al., 2015c] Park, H.-S., Lee, S., Son, S.-W., Feldstein, S. B., and Kosaka, Y. (2015c). The impact of poleward moisture and sensible heat flux on Arctic winter sea ice variability. J. Climate, 28:5030–5040.

[Woods and Caballero, 2016] Woods, C. and Caballero, R. (2016). The role of moist intrusions in winter Arctic warming and sea ice decline. J. Climate, 29:4473–4485.

2. I am not familiar with the details of the SOM method, and I am not illuminated by the description given in the text. You should give at least a concise description of the main idea behind SOM to give the reader some intuition into how to interpret the resulting patterns. I also do not understand why you start with 4x3=12 clusters and then subjectively group them in just 3 clusters. Isn't the point of clustering algorithms that they provide an objective classification? Why not just start with 3 clusters? More generally, why do you prefer SOM over alternatives such as k-means clustering?

Minor comment:

l.2 (Abstract): "It is assumed that through this decrease the large-scale circulation changes and therefore the meridional transport of heat and moisture increases". I have a hard time understanding this sentence. "It is assumed" by whom? What circulation changes are you referring to? Why should these changes lead to an increase in heat and moisture transport? The more natural assumption is that an increase in the heat transport leads to a decrease in the temperature gradient, not the other way around.
* * *

---

## Author Comment (AC1) · 28 Sep 2018

Dear anonymous reviewer #1,

we would like to thank the reviewer for the comments and ideas to improve the manuscript. Below we give some reply for part of the raised points, and will consider carefully all of them in the revised manuscript.

Indeed we used the ERA-Interim data on default pressure levels for vertical Integration. We will acquire the ERA-Interim data on model levels and will integrate the fields starting from there. Following the new calculations, we will clarify the description of the

calculations for the vertical integral.

Concerning the SOM method we used it because it omits linear assumptions for clustering the data. We will describe the process of grouping multiple patterns in more detail and we will consider showing the within cluster variance in the supplementary information. We consider showing the anomalies of the transport patterns. However, some of the patterns are very different and anomaly plots may show a similar distribution like the fluxes themselves. Temperature anomalies for each single pattern will be provided in the supplementary information, after the analysis based on model level data has been conducted.

The shown temperature anomalies were calculated from the winters from 1979-2016 mean. The presented standard deviation for the meridional transport has been computed from daily data. We will consider the statistical tests.

Our general aim was to link distinct directions of transport to the surface temperature. The reviewer raises the concern that no moisture flux has been considered. However, previous analyses have shown that the direction of the transport patterns are similar for moisture transports and the analyzed transports in this manuscript.

Concerning the reviewers idea of looking into vertical averaged / integrated temperature anomalies, we consider performing the analysis. Divergence of the heat flux patterns have not shown patterns that fit with the surface temperature anomalies in an obvious way. But we will consider repeating the analysis of the divergences with the model level data and the vertical averaged temperature anomalies.

The vertically integrated net meridional mass flux must be balanced, the compensation in Figure 4 is not that surprising. But the final result might change with the revised vertical integration.

We thank the reviewer for the other helpful comments, that will be taken into account for a revised version of the manuscript.

---

## Author Comment (AC2) · 28 Sep 2018

Dear Dr. Caballero,

we would like to thank you for the comments and ideas to improve the manuscript. We will carefully take them into consideration in a revised version of the paper.

The reviewer raises an interesting concern, when stating that the manuscript does not give much insight into physical mechanisms. However, our purpose was to investigate the possibility of using another method in analyzing the transport. Nevertheless, we will investigate the consequences for climate dynamics and discuss them in a more

comprehensive manner.

The goal of the work is focused just on the 'internal' heat transport, because we wanted to further clarify the concurrence of respective transports and surface temperatures. But, it is planned to redo calculations of the vertical integral (see answer to RC 1). Nevertheless, previous experiments have shown, that the general shape of the patterns are similar between different transports.

We will include more description on why and how we gathered the clustered data further. We chose SOM over k-means, because it has shown promising results in previous studies (e.g., Cassano et.al 2006). Further we think that the patterns emerging from the SOM are easier to relate to each other compared to the k-means. The SOM will inherently preserve the topology of the original data. However, for small numbers of clustered patterns the SOM behaves similar to the k-means. We will discuss this in more detail in revised version.

---

## Author Comment (AC3) · 10 Dec 2018

Response to the comments of reviewer #1,

We are thankful to the reviewer, whose comments helped us to improve the paper. We have revised the paper according the remarks, and hope that we sufficiently responded to each concern. In the following the reviewer´s concerns are repeated, and our respective responses is added in italics.

1. I am puzzled that there is such a strong heat transport over Greenland in many of the clusters (most pronounced in the cluster 2.2, 3.1 and 3.2). The transport clearly seems to increase there. I would expect the opposite since the vertical integral of the heat transport is proportional to the thickness of the layer from the surface to the 200 hPa level such that over high topography it should decrease. The same concerns intense transport over the Tibetean plateau and across the Rockies in some clusters. I strongly suspect that the authors used data on the default pressure levels for computng the vertically integrated heat transport and they did not take into account that some pressure levels intersect the surface. Over topography, where surface pressure can be below the pressure of, for example the 1000 hPa level, the ECMWF provides extrapolated fields. The heat flux computed from such extrapolated fields is unphysical.
This suspicion is further fuelled by the fact that there are no substantial T2m anomalies over Greenland in clusters with a strong heat transport there.[…]

*We repeated the calculations on model levels for the Moist Static Energy (MSE) instead of the internal Heat flux by using a ncl routine. However, the general transport structures remained similar compared to the previous analysis. All figures were changed for the MSE transport.*

2.The authors decide to perform a SOM analysis using 4x3 patterns. The number of patterns needs to be prescribed and is a subjective choice required by this method, which as such is fine. What I am worried about, however, is that they subsequently lump several clusters together according to the heat transport at 80°N, resulting in three major patterns, which are then considered for the rest of the paper. The only argument provided to justify this approach is that "this is common practice in SOM analysis". In my view, the SOM analysis on 2d fields is an overkill to obtain just these three patterns that are based on 1d fluxes at 80°N. If mainly the transport at 80°N is of interest for stratifying the patterns, why not perform an EOF or clustering analysis on the transport at 80°N and start from there? To make the effort put into the SOM analysis more valuable, I'd suggest to present some additional analyses that make use of the nuances displayed in the 12 SOM clusters. For example, the authors could display other composited fields for these clusters. These fields could, for example, be T2m anomalies, and frequencies of atmospheric weather systems (cyclones, blockings). The later would allow to relate the heat transports patterns to the dynamics, which I think would be a valuable addition and strengthen the paper.

*We included more explanation why it was decided to group patterns in the methods section 2.2 (P4L3ff). Indeed the phrasing was misleading concerning how the manual grouping was performed: we gathered the groups not based on true meridional transport into the central Arctic, but more generally with the general transports into mind.*

3.There is a misleading use of statistical significance testing at several occasions in the paper. Statistical significance testing provides information about the likelihood of e.g. observing a certain trend under the assumption of the null hypothesis (the trend is not real and just the

result of the sampling). It does not provide any evidence about whether the trend is real. A large p-value simply implies that the data is consistent with the null hypothesis and there is no evidence for a trend. It does also not rule out that there is a trend, but given the data we cannot tell. A low p-value in turn does not indicate a high likelihood for the hypothesis to be true, it just tells us that the data is unlikely to be observed under the assumption that the null hypothesis is true (for a thorough discussion see Ambaum 2010: Significance test in climate science, J. Climate, 23, 5927 – 5932).
Specifically in this paper the following misleading use of significance testing occurs:
∘P8 L6: "...the group of the Siberian pathway does not provide a significant trend..." This is a misleading statement, as it suggests that we should trust the weak trend in the frequency of the Siberian pathway less than the other trends, which we should trust because they are significant. This is of course absurd because the sum of the trends needs to balance. Hence, if we trust the trends of the other two patterns, we have to trust the weak trend of the third by as much. The significance test is therefore not helpful.
It would be more insightful to provide confidence intervals that illustrate the robustness of the trends to rule out that the trend is strongly influenced by a few data points.

*Thank you for this hint. We estimated the 95% confidence interval of the regression by applying bootstrap resampling. The Figure 6 was changed accordingly. The text passage at the end of section 3.4 (P8 L19ff) with the misleading phrasing was changed.*

∘P9 L11: "For the regions that correspond to lower temperatures with an increased occurrence of the North Atlantic Pathway and a decreased occurrence of the North Pacific Pathway no significant temperature trend can be found on the left panel of Fig. 7. This suggests that the temperature anomalies due to the transport changes are counteracted by other processes."
The reasoning is wrong here. It could well be that there is a cooling trend at these locations, but the trend is hidden because of one or two much warmer winters, which may be outliers. The statistical significance test does not provide us any information either way. A Monte Carlo resampling assessing the robustness of the local trends would give more insight.

*Thank you for this hint. We decided on estimating the slope with the Theil-Sen regression to be robust against outliers (P10L3).*

∘In Fig. 7 (left panel): The significance test here does again not provide any information about whether the trends in some regions are robust. Again confidence
intervals would be more insightful. Furthermore, in multiple testing scenarios, if any significance test is done at all, a field significance test should be done to take spatial correlations and erronous rejections of the null hypothesis into account (cf. Ventura et al. 2004: Controlling the Proportion of Falsely Rejected Hypotheses when Conducting Multiple Tests with Climatological Data, J. Climate, 17, 4343 – 4356)

*We decided to omit any significance tests for this figure, because significance for the temperature trends are not helpful for the conducted comparison and have led to misleading interpretations. We changed the text passages in section 3.5 accordingly.*

4.What is the reason for not considering moisture fluxes as well? These are arguably highly important for Arctic heat anomalies because of their impact on the radiation balance.
See for example Woods et al. (2013) and Messori et al. (2018; cited in the paper).
Woods, C., R. Caballero, and G. Svensson (2013), Large-scale circulation associated with moisture intrusions into the Arctic during winter, Geophys. Res. Let.,40, 4717–4721,

doi:10.1002/grl.50912.

*Thank you for this suggestion. We changed our calculations to using the Moist Static Energy to also consider moisture flux. All figures were redone accordingly.*

5.For displaying the differences between individual clusters more clearly, it could help to show heat transport anomalies instead of the full transports. In many patterns the differences in the heat transport are rather nuanced and hard to see.

*We decided not to show the anomalies of the transports for the SOM clusters. We hope that the clusters are distinguishable enough. Differences are sometimes not easy to digest, because the difference arrows would point into very different directions than the actual fluxes. Further, we think that the presentation of the clusters is sufficient because we anyhow want to focus our analysis on the three pathways.*

6. How large is the within cluster variance for the SOM clusters and the three main clusters?

*The mean within cluster variances for the three main clusters is about 2.65e22. While it is about 2.2e22 for the 12 SOM clusters. The numbers were added to section 3.1(P4L32f).*

7.P5 L10f: T2m anomalies: Why don't you consider vertically averaged / integrated (potential) temperature anomalies? These would be more clearly related to the heat flux divergence than T2m anomalies, which are strongly influenced by surface heat fluxes. This is especially true in regions with a rapidly declining wintertime sea ice cover (Barents and Kara Seas), where the temperature trends are to a large extent due to surface heat fluxes.

[Figure]

*Figure 4: Composite of vertically integrated potential temperatures minus mean for the analyzed time frame.*

*Figure 4 shows the vertically integrated composite minus mean of the potential temperature. Generally the Pacific Pathway shows negative anomalies over the whole Arctic, while the Atlantic Pathway shows positive Anomalies for Eurasia and the Barents, Kara, and*

*Laptev sea.The Figure 4 was added, described, and compared to the surface air temperature anomalies in section 3.2(7L7ff)*. A short paragraph was included in the discussion (P11L19)

8.Related to the above I would be interested in seeing the divergence of the heat flux. Warming at a certain location will be more related to the heat flux divergence than the flux itself.

[Figure]

*Figure A: Composite of mean of vertically integrated MSE transport divergence minus mean of the analyzed time frame.*

> *Fig. A shows the anomaly of the composite of the vertically integrated divergence of the horizontal MSE transport. The major differences occur over Greenland compared to the Arctic ocean. We decided not to show the divergence as the main differences are in regions with high topography, and do not provide useful information for large part of the Arctic regions. An explanation was added in section 3.2 (P8L1f)*

9.P5 L10f: How are anomalies computed? Are they taken from the period (DJF 1979 – 2016) mean or is a running mean used to account for intra-seasonal variations?

> *Anomalies were calculated from the mean of the given period. We added a description in the beginning of section 3.2(P5L1).*

10. Fig. 4: There seems to be a large compensation between the poleward and the equatorward transports, which I find surprising, especially concerning the strong southward heat transport at -120°E, which must be associated with very cold air (with low heat content).
And does the standard deviation depict the inter-annual variability? That is, is it computed from the means of each winter? Or is it the standard deviation computed from daily data?

> *We do not show advection so the southward transport anomaly's amplitude is either controlled by the transport of warmer air into the south, or very strong southward winds. We added a short clarification (P8L8).*
> *We calculated the standard deviation from daily data and added the respective description in the figure 5 caption (P8 Fig 5).*

11. Fig. 7 (right panel): The caption should state that for the North Pacific pathway the

inverse of the temperature anomaly was taken (as described in the text).
*Done.*

12. P1 L16: To first order the much stronger warming in the Arctic compared to lower latitudes is caused by the loss of sea ice, exposing major areas of the Arctic ocean to the atmosphere, leading to subsequent warming of the lower troposphere, and not the other way round. Additional melting of sea ice because of Arctic amplification would require additional transport of heat into the Arctic.
*We changed the passage so it is hopefully less confusing/misleading (P1L17f).*

13. P2 L1: "To summarize, there is a clear indication that Arctic Amplification alters the circulation and heat transport patterns in the Arctic." I'd suggest to tone this statement down a bit. The causality is not fully clear in my view. See also Screen et al. 2018: Consistency and discrepancy in the atmospheric response to Arctic sea-ice loss across climate models, Nat. Geosci., 11, 155 – 164
*We eased the tone of the statement(P2L1).*

14. P2 L6: "... that have been emerged ..." → "... that have emerged ..."
*Done.*

15. P2 L9: pattern should be patterns
*Done.*

16. P2 L9: either high or strong, not both
*Done.*

17. P2 L26: reanalyses → reanalysis
*Done.*

18. P2 L28: Please rephrase "This is used to obtain informations from the whole tropospheric column."
*We deleted this phrase.*

19. P3 L11: "... an average picture ..." → the heat transport throughout the entire troposphere (?)
*Done.*

20. P4 L4: Fig. → Figs.
*Done.*

21. P4L6: This is likely an artefact from the vertical averaging
*These transports have also emerged from the data based on level data without extrapolated fields over high altitude regions.*

22. P5 L1: ... are directed → ... is directed …
*Done.*

23. P5 L4: zonally → zonal

*Done.*

24. P5 L7: Awkward formulation " ... with two cyclone motions ...", please rephrase
   *Done.*

25. P5 L9: "... an ant-cyclone motion..." dito
   *Done.*

26. P6 L12: Why focus on 75°N when SOM clusters are grouped together according to the heat flux at 80°N? Generally, I think 75°N is better suited because 80°N lies largely in the interior Arctic (except for the European sector).
   *We clarified the statement concerning the grouping of SOM cluster (P4L14).*

27. P6 L33: Since you integrate H vertically, you could simply state that you consider the meridional component of the heat flux Eq. (1).
   *Done.*

28. P7 L4: Remove "Generally, the meridional transports of the three groups ft well to the described pathways." - of course they have to be consistent as you look at the same quantity (the vertically integrated heat transport).
   *Done.*

29. P8 L2: occur → occurs
   *Done.*

30. P9 L14: remove "can not"
   *Done.*

31. P10 L6: favors → favor
   *Done.*

32. P10 L20: measurement → measurements
   *Done.*

33. P10 L25: Awkward phrasing, please rephrase.
   *Done.*

34. P10 L28: the presented work here → the work presented here
   *Done.*

35. P10 L31: a increase → an increase
   *Done.*

36. P11 L3: generally → general
   *Done.*

37. P11 L13: that at region → that in regions
   *Done.*

38. P11 L16: changing of → changes in
   *Done.*

39. P11 L20f: awkward phrasing until and including "... whole picture."
    *Done.*

40. P11L27: an guide → a guide
    *Done.*

---

## Author Comment (AC4) · 10 Dec 2018

Dear Dr. Caballero,

We are thankful for your comments that very much helped us to improve the paper. We have revised the paper according to the remarks, and hope that we sufficiently responded to each concern. In the following your concerns are repeated, and our respective responses is added in italics.

1. The study claims to study "heat transport", but actually only studies one component of the heat transport. The relevant quantity for atmospheric energy transport is the moist static energy, h = c_p T + g z + L_v q (where g is gravitational acceleration, z geopotential height, L_v latent heat of vaporization and q specific humidity). The authors only consider the first term, and neglect the others for no clear reason. In fact, recent work (see references below) shows that the latent heat component (i.e. the moisture transport) is the most important for warming the surface in the Arctic. The authors should cite these papers. Even the Yoshimori et al. paper, which is cited by the authors, makes this point very clearly. The fact that moisture transport is not considered makes physical interpretation of the authors' results difficult – it's not clear if there is any direct causality implied by the relation between sensible heat transport and surface temperature anomalies shown here. It is thus not clear to me how this paper contributes to the current debate about Arctic warming. To make a clear and useful contribution, the authors really would need to apply their SOM classification to moisture transport and assess the pathways they obtain. It would also be useful to do a classification for dry static energy (c_p + g z) transport.
[Lee et al., 2017] Lee, S., Gong, T., Feldstein, S. B., Screen, J. A., and Simmonds, I. (2017). Revisiting the cause of the 1989–2009 Arctic surface warming using the surface energy budget: Downward infrared radiation dominates the surface fluxes. Geophys. Res. Lett., 44:10,654–10,661.
[Park et al., 2015a] Park, D.-S. R., Lee, S., and Feldstein, S. B. (2015a). Attribution of the recent winter sea ice decline over the Atlantic sector of the Arctic Ocean. J. Climate, 28:4027–4033.
[Park et al., 2015b] Park, H.-S., Lee, S., Kosaka, Y., Son, S.-W., and Kim, S.-W. (2015b). The impact of Arctic winter infrared radiation on early summer sea ice. J. Climate, 28:6281–6296.
[Park et al., 2015c] Park, H.-S., Lee, S., Son, S.-W., Feldstein, S. B., and Kosaka, Y. (2015c). The impact of poleward moisture and sensible heat flux on Arctic winter sea ice variability. J. Climate, 28:5030–5040.
[Woods and Caballero, 2016] Woods, C. and Caballero, R. (2016). The role of moist intrusions in winter Arctic warming and sea ice decline. J. Climate, 29:4473–4485.

*Thank you for this suggestion. We now changed our analysis to the Moist Static Energy (MSE) transport and repeated all the calculations. The general transport structures remained similar compared to the previous analysis.*

2. I am not familiar with the details of the SOM method, and I am not illuminated by the description given in the text. You should give at least a concise description of the main idea behind SOM to give the reader some intuition into how to interpret the resulting patterns. I also do not understand why you start with 4x3=12 clusters and then subjectively group them in just 3 clusters. Isn't the point of clustering algorithms that they provide an objective classification? Why not just start with 3 clusters? More generally, why do you prefer SOM over alternatives such as k-means clustering?

*We added some more description to the SOM method in the beginning of section 2.2 (P3L16ff). We added an explanation why we grouped data and why SOM were chosen over k-means at the end of section 2.2(P4L3).*

Minor comment:
l.2 (Abstract): "It is assumed that through this decrease the large-scale circulation changes and therefore the meridional transport of heat and moisture increases". I have a hard time understanding this sentence. "It is assumed" by whom? What circulation changes are you referring to? Why should these changes lead to an increase in heat and moisture transport? The more natural assumption is that an increase in the heat transport leads to a decrease in the temperature gradient, not the other way around.

*We slightly modified the Abstract.*

---

## Referee Report (RR1)

Review of ACPD-2018-636 – revision 1

**"Heat Transport Pathways into the Arctic and their Connections to Surface Air Temperatures"**

by D. Mewes and C. Jacobi

The authors have extended their analyses to include moisture transport by computing vertically integrated moist static energy fluxes. Accordingly they have redone most of the analyses. In particular, they have fixed the vertical integration by using data on model levels. Furthermore, they have addressed many of my concerns, including those regarding significance testing, leading to an overall improved manuscript. Nevertheless, I still have two major reservations. These are:

- The grouping of the clusters into three groups is not sufficiently explained and it remains unclear what the criteria are; see points (1) and (2) below.
- The patterns of 2m temperature or vertically integrated potential temperature anomalies and the flux of moist static energy do not match very well and their linkage is, therefore, not clear based on the analyses presented. In order to make a more convincing case, this must be supported by additional analyses or the origin of the differences between the patterns must be explained in more detail.

In my view these issues are critical and must be addressed before the paper can be accepted for publication.

**Specific comments:**

1.  Thank you for the explanation of the grouping of SOM clusters. However, I still find it somewhat difficult to understand what the specific criteria for the grouping are. I assume that by "general transports", as stated in the response, you refer to the 2d vector fields of the energy transports. If so, what are the specific characteristics of the general transports that inform the grouping?

2.  Furthermore, the following statement is hard to understand:

    P4L5: "The mathematical description of the Euclidean distance might assign distinct fields to patterns that fit mathematically but not under a meteorological point of view."

    What distinct fields does the mathematical description of the Euclidian distance assign to patterns? Maybe the authors want to say, that, in addition to grouping clusters whose mutual Euclidian distance is small compared to the other clusters, also other meteorological criteria could be used. If so, what are these meteorological criteria specifically?

3.  Thank you for computing vertically integrated potential temperature anomalies and the divergence of the of the MSE fluxes. I find it difficult to relate these patterns to the MSE

fluxes and the 2m temperature anomalies. More explanation should be given how this linkage.

The missing link is probably contained in the divergence of the MSE fluxes. From the figures provided, these cannot be related due to the strong divergence / convergence along orography (which is likely an artefact from the numerical computation?).

4. Please include letters (a), (b), ... for panels in Figures and use these to refer to the inidvidual panels in the text (instead of e.g., "right" panel of Fig. XYZ).

5. P1L17: please rephrase "... and following that ... " → " ... and the consequent ... " (?)
6. P3L18: typo "build" → "built"
7. P5L3: typo "deviation" → "deviations"
8. P7L17: typo "dos" → "does"
9. P11L23: typo "frequent" → "frequently"
10. Figures: refer to panels by a, b, c...
11. P12L9: Awkward phrasing, please rephrase: "... presented by the pathways presented in this work:"
12. P12L12: typo "latent heat transport only."
13. P12L16: "which shows to have a positive trend" → "which features a positive trend"?
14. P12L17: "In connection, ... " awkward phrasing, please rephrase this sentence.

---

## Author Response (AR2)

Dear anonymous reviewer #1,

We are thankful for your comments that very much helped us to improved the paper. We have revised the paper according to the remarks, and hope that we sufficiently responded to each concern. In the following your concerns are repeated, and our respective responses are added in italics.

**"Heat Transport Pathways into the Arctic and their Connections to Surface Air Temperatures"**

by D. Mewes and C. Jacobi

The authors have extended their analyses to include moisture transport by computing vertically integrated moist static energy fluxes. Accordingly they have redone most of the analyses. In particular, they have fixed the vertical integration by using data on model levels. Furthermore, they have addressed many of my concerns, including those regarding significance testing, leading to an overall improved manuscript. Nevertheless, I still have two major reservations. These are:

- The grouping of the clusters into three groups is not sufficiently explained and it remains unclear what the criteria are; see points (1) and (2) below.

- The patterns of 2m temperature or vertically integrated potential temperature anomalies and the flux of moist static energy do not match very well and their linkage is, therefore, not clear based on the analyses presented. In order to make a more convincing case, this must be supported by additional analyses or the origin of the differences between the patterns must be explained in more detail. In my view these issues are critical and must be addressed before the paper can be accepted for publication.

*Thank you for these concerns. We have now explained the grouping in more detail, and recalculated the averaged potential temperature, which unfortunately had been erroneous in the earlier version,*

Specific comments:
1. Thank you for the explanation of the grouping of SOM clusters. However, I still find it somewhat difficult to understand what the specific criteria for the grouping are. I assume that by "general transports", as stated in the response, you refer to the 2d vector fields of the energy transports. If so, what are the specific characteristics of the general transports that inform the grouping?
  *The specific characteristics were mainly the direction and rotation of the vector fields. For example the patterns 1.1 and 1.2 show similar meridional transports through the Fram Strait and similar rotational features west of Greenland.*
*Patterns 3.1 and 3.1 both show distinct rotation north of Siberia, while both feature most fluxes from North Siberia to North America.*
*Patterns 3.3 and 3.4 are fit for grouping because they share very similar rotational centers North of Siberia and West of Greenland while showing distinct meridional transports through the Bering Strait.*
*We specified the grouping characteristics in the results and in the description of the SOM.*

2. Furthermore, the following statement is hard to understand:
P4L5: "The mathematical description of the Euclidean distance might assign distinct fields to patterns that fit mathematically but not under a meteorological point of view."

What distinct fields does the mathematical description of the Euclidian distance assign to patterns? Maybe the authors want to say, that, in addition to grouping clusters whose mutual Euclidian distance is small compared to the other clusters, also other meteorological criteria could be used. If so, what are these meteorological criteria specifically?

> *We clarified the explanation concerning the Euclidean distances and specified the meteorological criteria in the description of the SOM.*

3. Thank you for computing vertically integrated potential temperature anomalies and the divergence of the of the MSE fluxes. I find it difficult to relate these patterns to the MSE fluxes and the 2m temperature anomalies. More explanation should be given how this linkage.
The missing link is probably contained in the divergence of the MSE fluxes. From the figures provided, these cannot be related due to the strong divergence / convergence along orography (which is likely an artefact from the numerical computation?).

> *Generally, these linkage is not easy to see within the vertically integrated fluxes and the surface temperature. Like described in Graversen (2016), the maximum influence of the transports on the surface is delayed by about 5 days. Hence, we have not considered fully the persistence of each pathway. Right now we are not able to distinguish the 'real' resulting temperatures due to the MSE flux. It is planned for future analyses to take these delayed responses into account.*
> *However, within this study we put the focus on a snapshot of the atmosphere under these distinct MSE fluxes to get a broad idea of the 2 meter temperature during the respective patterns.*
> *We added a paragraph which describes this in the Discussion.*
> *Concerning the potential temperature we checked the data and plotting again and found an error in the calculations by not taking into account the whole data base but only a single year. We apologise for this mistake. We fixed the problem and included the new results.*
> *The new results for the vertically averaged potential temperature anomaly composites show now very good agreement with the 2 meter temperature anomaly composites. We can conclude that the 2 meter temperature anomaly composites represent the general tropospheric state well.*

4. Please include letters (a), (b), ... for panels in Figures and use these to refer to the inidvidual panels in the text (instead of e.g., "right" panel of Fig. XYZ).

> *Done*

5. P1L17: please rephrase "... and following that ... " → " ... and the consequent ... " (?)

> *Done*

6. P3L18: typo "build" → "built"

> *Done*

7. P5L3: typo "deviation" → "deviations"

> *Done*

8. P7L17: typo "dos" → "does"

> *Done*

9. P11L23: typo "frequent" → "frequently"

> *Done*

10. Figures: refer to panels by a, b, c…

> *Done*

11. P12L9: Awkward phrasing, please rephrase: "... presented by the pathways presented in this work:"
   *Done.*

12. P12L12: typo "latent heat transport only."
   *Done*

13. P12L16: "which shows to have a positive trend" → "which features a positive trend"?
   *Done*

14. P12L17: "In connection, ... " awkward phrasing, please rephrase this sentence.
   *Done*

Graversen, R. G.: Do changes in the midlatitude circulation have any impact on the Arctic surface air temperature trend?, Journal of climate,
19, 5422–5438, https://doi.org/10.1175/JCLI3906.1, 2006

Dear Prof. Dr. Caballero,

We are thankful for your comments that very much helped us to improved the paper. We have revised the paper according to the remarks, and hope that we sufficiently responded to each concern. In the following your concerns are repeated, and our respective responses are added in italics.

The authors have responded well to comments in the first review, and I find the current version to be very much improved. It provides a set of interesting results which are consistent with previous work but obtained using different methodology, providing mutual reinforcement. I think the current version can be published subject to some very minor revisions listed below.

Fig 4: Please state over which pressure range the pot. temperature is integrated. The units would be easier to interpret if you showed vertically *averaged* rather than integrated pot. temperature (pattern should not change)

*We added the description of the pressure range of the averaging in the text and in the caption. We changed to plot to show the average. During this process we found errors in our calculations, and included the new results. These new results now fit the 2 meter temperature anomalies very well.*

p4 l.1: "manually" written twice, one is redundant

*The first "manually" was removed.*

p4 l.23: origins from -> originates in

*Done.*

p10 l.9: I think rather than "inverse", you mean "the negative of"?

*Additionally we changed the text in the caption of the respective Figure.*

[revised manuscript text omitted]

---

## Author Response (AR3)

Dear Prof. Dr. Wernli,

We thank you as the editor, and the reviewers for your time and valuable remarks. We have revised the paper according to your remarks, and hope that we sufficiently responded to each remark. In the following your concerns are repeated, and our respective responses is added in italics.

Dear Mr. Mewes
Many thanks for your revisions. I now also had a detailed look at your revised paper. I appreciate the main result, that parts of the Arctic temperature trend can be attributed to changes in large-scale transport patterns (Fig. 8). The paper is now almost ready for publication, however, some changes to the language are still required and a few important clarifications would be very helpful for the future readers of your study (see comments below). Please very carefully check your revised version for typos etc.
With best regards,
Heini Wernli
*Thank you again for your comments that helped us improve the manuscript.*

p. 1 line 17 and in other places: "Arctic Amplification" --> "Arctic amplification"
*Done.*

p. 2 line 8: "Aleuten low" --> "Aleutian Low"
*Done.*

p. 3 line 23: I don't understand the sentence "Thereby ..." can it be omitted or can you reformulate it and make it more understandable?
*The sentence can be omitted and has been removed*

p. 3 line 27: "patterns emerges" --> "patterns emerge"
*Done.*

p. 3 line 29: you never explain what "Eucledian distance" (and in what region, only Arctic or entire Hemisphere?) you are using for your SOM analysis, please explain! The description should be such that the results become reproducible. Also further below (p. 4 line 8) you write "due to the characteristics of Euclidean distance" but the reader does not know what this distance is!
*We included an additional explanation, which two fields has been used to calculate the Euclidean distance.*

p. 4 lines 9 and 13: "under" --> " from a"
*Done.*

p. 4 line 15: I don't know what you mean with "intrinsic topology of the data"
*We changed the statement to be more descriptive.*

p. 4 line 20: "Arctic Ocean" --> "Arctic"
*Done.*

p. 4 line 25 and throughout the paper: I suggest that you write "Pathway" with a small "p"
   *Done.*

p. 4 line 28: "from North Atlantic" --> "from the North Atlantic"
   *Done.*

p. 4 line 34: "centralized center" ?? just "center" is fine
   *Done.*

Caption of Fig. 1: please explain clearly that vectors show MSET, and colors the magnitude of MSET (is this correct?).
   *Yes this is correct, we included the explanation in the caption.*

p. 6 line 3: "of the temperature anomalies" --> "of the 2-m temperature anomalies"
   *Done.*

p. 6 line 4: you can delete the sentence "The red framed plot ..." since this is clear from the figure caption.
   *Done.*

Caption of Fig. 3: "for each of three the pathways" --> "for the three pathways"
   *Done.*

Caption of Fig. 4: "hPA" --> "hPa"
   *Done.*

p. 8 line 1: "a elongated" --> "an elongated"
   *Done.*

p. 8 line 7: you just show Fig. 5 but don't comment on it. What should the reader learn from this figure? Maybe Figs. 5 and 6 could be combined into one figure with 4 panels in a 2x2 arrangement?
   *We decided on combining the two Figures.*

Caption of Fig. 7: "for each years winter" --> "for each winter"
   *Done.*

p. 8 line 9: I don't understand, what do you mean by "No advection (transport across the gradient) is shown"; and why "No" with capital N after ";"?
   *This part is not helpful and don't provide additional information. We removed it.*

Fig. 8: Is this for 2-m temperature or for the vertical integral, please specify!
   *We renamed the whole section 3.5 to '2-m temperature trends' and changed the caption of Figure 8 accordingly.*

p. 10 line 16: "are broadly inverse to each other" --> "are typically of opposite sign"
   *Done.*

p. 10 line 18: "then" --> "than"
   *Done.*

p. 11 line 6: "It has to be noted, that the MSE transport cannot be accounted for changes in the

temperature anomalies alone." This sentence has grammatical problems. Most likely you like to say "Please note that the changes in MSE transport cannot account for the entire temperature trends."

*We changed the sentence according to your suggestion.*

p. 11 line 12: "It has to be noted that the linkage ... is not straightforward."

*Done.*

p. 11 line 14: I find it confusing that you mention here that MSE flux convergence and T2m did not show good agreement, because you never showed this field?? Also, why do you write here about MSE fluxes and in the rest of the paper about MSE transport?? Please clarify.

*We removed the sentence concerning the MSE transport convergence.*

p. 11 line 21: why "and"?? (4th word), I think this can be deleted

*We deleted the "and".*

p. 15 line 22: "climate" --> "Climate"

*Done.*

List of references: for most references, you write the full journal names, but for some you use abbreviations. Please make this consistent.

*We changed added the abbreviations for the journal names for all references.*

[revised manuscript text omitted]